# The Effects of Human BDH2 on the Cell Cycle, Differentiation, and Apoptosis and Associations with Leukemia Transformation in Myelodysplastic Syndrome

**DOI:** 10.3390/ijms21093033

**Published:** 2020-04-25

**Authors:** Wen-Chi Yang, Sheng-Fung Lin, Shu-Chen Wang, Wan-Chi Tsai, Chun-Chieh Wu, Shih-Chi Wu

**Affiliations:** 1Division of Hematology and Medical Oncology, Department of Internal Medicine, E-DA Hospital, Kaohsiung 824, Taiwan; shlin@kmu.edu.tw; 2Faculty of School of Medicine, College of Medicine, I-Shou University, Kaohsiung 824, Taiwan; 3Department of Laboratory Medicine, Kaohsiung Medical University Hospital, Kaohsiung 807, Taiwan; sjwang@cc.kmu.edu.tw; 4Department of Medical Laboratory Science and Biotechnology, Kaohsiung Medical University, Kaohsiung 807, Taiwan; wanchi@kmu.edu.tw; 5Department of Pathology, Kaohsiung Medical University Hospital, Kaohsiung 807, Taiwan; lazzz.wu@gmail.com; 6School of Medicine, China Medical University, Taichung 404, Taiwan; rw114@mail.cmuh.org.tw; 7Trauma and Emergency Center, China Medical University Hospital, Taichung 404, Taiwan

**Keywords:** acute leukemia, BDH2, LCN2, myelodysplastic syndrome (MDS), iron metabolism

## Abstract

Iron overload is related to leukemia transformation in myelodysplastic syndrome (MDS) patients. Siderophores help to transport iron. Type 2-hydroxybutyrate dehydrogenase (BDH2) is a rate-limiting factor in the biogenesis of siderophores. Using qRT-PCR, we analyze *BDH2*mRNA expression in the bone marrow (BM) of 187 MDS patients, 119 de novo acute myeloid leukemia (AML) patients, and 43 lymphoma patients with normal BM. Elevated *BDH2*mRNA expression in BM is observed in MDS patients (*n* = 187 vs. 43, normal BM; *P* = 0.009), and this is related to ferritin levels. Patients with higher *BDH2* expression show a greater risk of leukemia progression (15.25% vs. 3.77%, lower expression; *P* = 0.017) and shorter leukemia-free-survival (medium LFS, 9 years vs. 7 years; *P* = 0.024), as do patients with a ferritin level ≥350 ng/mL. Additionally, we investigate the mechanisms related to the prognostic ability of BDH2 by using BDH2-KD THP1. The cell cycle analysis, surface markers, and special stain studies indicate that BDH2-KD induces differentiation and decreases the growth rate of THP1 cells, which is associated with the retardation of the cell cycle. Moreover, many genes, including genes related to mitochondrial catabolism, oncogenes, tumor suppressor genes, and genes related to cell differentiation and proliferation influence BDH2-KD THP1 cells. Herein, we demonstrate that BDH2 is involved in cell cycle arrest and the inhibition of differentiation in malignant cells. Furthermore, the high BDH2 expression in MDS patients could be suggestive of a poor prognostic factor. This study provides a foundation for further research on the roles of BDH2 and iron metabolism in the pathogenesis of MDS.

## 1. Introduction

Myelodysplastic syndrome (MDS) is a heterogeneous group of clonal hematopoietic stem cell disorders characterized by ineffective hematopoiesis, leading to unilineage and multilineage cytopenias and dysplasia. This disease can progress from cytopenia(s) to acute myeloid leukemia (AML) through several intermediate subtypes [1,2,3,4,5]. Although several genes and splicing pathway mutations, as well as cytogenetic changes, have been reported to be associated with MDS, including mutations of *RAS*, *TP53*, *RUNX1*, *ASXL1*, *c-CBL*, *DNMT3A*, *TET2*, *EZH2*, *U2AF1*, *U2AF35*, *ZRSR2*, *SRSF2*, and *SF3B* [6,7,8,9,10], the genetic changes associated with the pathogenesis of MDS still remain unclear.

Anemia resulting from multiple blood transfusion induced iron accumulation [11,12] or related to growth differentiation factor-11 (GDF11), GDF15, and hepcidin [13,14,15] is one of the characteristics of MDS [16]. Excess iron in MDS patients is associated with multiple organ damage and is responsible for an increased leukemia transformation rate [14,17], as well as shortened leukemia-free survival (LFS) and overall survival (OS) [18,19].

Lipocalin (LCN2) 24p3 is an iron-trafficking protein that requires small-molecular-weight iron-chelating compounds to sequester iron [20,21]. Devireddy et al. reported that the 24p3-associated mammalian siderophore 2,5-dihydroxybenzoic acid (2,5-DHBA) [22,23] is catalyzed by the enzyme cytosolic type 2-hydroxybutyrate dehydrogenase (BDH2) [23,24] and is related to LCN2 24p3-mediated iron transport and apoptosis [23]. The key physiological implication of BDH2 is that iron-mediated post-transcriptional regulation of human BDH2 controls mitochondrial iron homeostasis in human cells [25]. We observed that BDH2 expression is an independent poor prognostic factor for cytogenetically normal AML (CN-AML), as it plays an anti-apoptotic role [26]. In the present study, we investigated whether BDH2 can serve as a prognostic marker for MDS and act as a predictor for the progression of leukemia. Furthermore, we used THP1, an acute myelomonocytic leukemia cell line, to present the possible mechanism of BDH2-related leukemia transformation in vitro. The THP1 cell line has been used for MDS and AML research in many fields [27,28,29].

## 2. Results

### 2.1. Patient Characteristics

We enrolled 318 patients, including 199 newly diagnosed MDS patients and 119 de novo AML patients, at Kaohsiung Medical University, Chung-Ho Memorial Hospital, Taiwan, from 2001 to 2012, and they were reviewed until the end of 2019. We also enrolled 40 normal controls. The characteristics of patients are shown in Table 1. A total of 187 MDS patients with good mRNA quality were examined, including 114 patients with low BDH2 mRNA expression (BDH2^Low^) and 73 patients with high BDH2 mRNA expression (BDH2^High^). The patients in both groups were well-matched for age and gender. Patients were classified based on World Health Organization (WHO) criteria and Revised International Prognostic Scoring System (IPSS-R) scores. The patients in the MDS, de novo AML, and normal BM control groups were well-matched with regard to gender distribution. The median ages of patients with MDS, de novo AML, and normal BM were 64.47 (19–88), 60 (21–88), and 55 (32–65) years old, respectively.

### 2.2. Expression of BDH2 and LCN2 in MDS and Control Patients

The expression of *BDH2*mRNA was significantly elevated in MDS patients, especially those who were high risk (RAEB-1and RAEB-2), compared to patients with normal bone marrow (BM) (*P* = 0.009). Further, the expression of *BDH2*mRNA was higher in de novo AML patients than in MDS patients (*p* < 0.001; Figure 1A). Conversely, *LCN2*mRNA expression was significantly lower in MDS patients compared with the normal population, and it reduced further with an increase in blast cells. *LCN2*mRNA expression levels were significantly lower in de novo AML BM samples than in those of MDS patients and the normal population (*p* < 0.001; Figure 1B). It was also noted that *LCN2*mRNA expression was higher in patients with low *BDH2* expression (*P* = 0.015; Appendix A). However, no significant correlation was observed between *BDH2* and *LCN2* mRNA expression levels in the BM of MDS patients (*P* = 0.816; Appendix A). According to the IPSS-R prognostic scores, *BDH2*mRNA expression levels were higher in high- and very high-risk MDS patients than in low-risk patients (Table 1).

We analyzed 13 patients using BM samples preserved at different stages of MDS. Of these, four patients showed increases in *BDH2* mRNA expression under progress. Others showed a mild decrease or no change in *BHD2* mRNA expression during disease progression (Appendix A).

### 2.3. BDH2 Expression Is Related to Ferritin Levels

Ferritin levels were available for 115 MDS patients at time of diagnosis before they underwent repeated blood transfusions. Plasma ferritin levels showed no significant linear relationship with the mRNA expression levels of *BDH2* and *LCN2* (Appendix A) However, we observed that serum ferritin levels were higher in patients belonging to the high *BDH2*mRNA expression group (*P* = 0.02; Table 1 and Appendix A).

### 2.4. Prognosis, BDH2 Expression, and Ferritin

Patients in the BDH2^High^ group showed a higher risk (15%) for leukemia progression during a long-term follow-up period than patients in the BDH2^Low^ group (3.81%) (*P* = 0.017; Table 1). Further, patients in the BDH2^Low^ group had a longer LFS than those in the BDH2^High^ group, with medium LFS of 9 and 7 years, respectively (*P* = 0.024; Figure 2A), but they did not have a longer OS (*P* = 0.121; Figure 2B). It is noteworthy that the leukemia transformation rate is related to subtypes of MDS (refractory anemia with ringed sideroblasts (RARS): 0%; refractory anemia (RA): 2%; refractory anemia with excess blasts (RAEB)1: 11.11%; RAEB2: 20.51%; myelodysplastic syndrome/myeloproliferative neoplasm (MDS/MPN): 8.33%; *P* = *0.002*). We found that the leukemia transformation rate was higher in the BDH2^High^ group in each subtype, except for RARS and MDS/MPN patients (BDH2^High^ vs. BDH2^Low^ group: 0 vs. 0 in RARS; 6.25% vs. 0 in RA; 25% vs. 0 in RAEB1; 20% vs. 21.05% in RAEB2; 0 vs. 9.09% in MDS/MPN).Greenberg et al. defined the ferritin level as a survival predictor with a cut-off value of 350 ng/mL [30]. Patients with serum ferritin <350 ng/mL had a trend toward having a longer LFS and significantly improved OS, with medium OS of 8.8 years and LFS of 2.6 years, respectively (Figure 2C,D).

### 2.5. BDH2 Knockdown (KD) THP1 Showed a Slow Growth Rate Related to Cell Differentiation and Cell Cycle Retardation

#### 2.5.1. Cell Cycle Retardation

The expression levels of *BDH2*mRNA and *LCN2*mRNA in each shRNA BDH2 interference THP1 cell line using qRT-PCR are shown in Appendix A. The strain with the greatest KD efficiency in both mRNA (Appendix A) and protein levels (Appendix A) was shRNA-BDH2-3 THP1. We analyzed the growth rate after starving native THP1 cells, THP1 transfected with an shRNA empty vector, and shRNA-BDH2 THP1 cells in serum-free medium. The regrowth rate after reculture in serum-containing medium was retarded in BDH2-KD cells and showed dependency on *BDH2* expression levels (Figure 3A).

#### 2.5.2. Increased Differentiation in shRNA-BDH2 THP1 Cells

To study the reason for growth retardation, the surface markers CD11b, CD14, CD15, CD16, and CD64 were analyzed by flow cytometry. It was found that CD11b and CD15 expression increased in shRNA-BDH2 THP1 cells under vitamin D3 treatment, while the difference in CD15 expression was not significant (Figure 3B). There was no difference in CD14 and CD16 expression (Appendix A). CD64, which is expressed in both immature and mature monocytic series cells, was not influenced by BHD2 (Appendix A). In the special stain analysis, a slight increase in the myeloperoxidase (MPO) positive stain and a significant increase in the nonspecific esterase (NSE) stain in BDH2-KD cells were observed; these changes were related to the mRNA expression levels of *BDH2* (Figure 3C). Further, during Liu staining for the morphological analysis, it was noticed that the concentration of macrophage-like cells increased in the BDH2-KD group (Figure 3D).

#### 2.5.3. Longer Cell Cycle Arrest in shRNA Interference BDH2-KD THP1 Cells

After starvation and restart of the cell cycle, we noticed that while native and shRNA control vector transfected THP1 cells started the synthesis phase and reached a plateau from day 3, shRNA-BDH2 THP1 cells started synthesis late at day 4 and continued to show increased synthesis up to day 7 (Figure 4 and Appendix A).

### 2.6. BDH2 in Myeloid and Erythroid Series

To detect BDH2 protein expression in BM, immunohistochemistry was performed. The results showed that both erythroid and myeloid series were positive for BDH2. Immature cells such as blast cells, and mature cells such as band and segmented cells, stained positive for BDH2 antibodies (Appendix A).

### 2.7. BDH2 Targets on Multiple Genes Related to Cell Proliferation, Survival, and Cancer Pathway

Under microarray study, many genes related to cell proliferation, growth and survival, and cancer, including colon cancer, pancreatic cancer, and acute leukemia, showed up- or downregulation in BDH2-KD THP1 cells (Figure 5 and Appendix A, Appendix A). *TGFBR1* upregulation, *SMAD3* downregulation, *MAPK9* upregulation, and *TP53INP2* downregulation can explain stress-induced apoptosis. The phenomenon of *TNFSF13B*, *MMP9*, *PI3KCD*, and *TP53INP2* downregulation may be related to poor proliferation of BDH2-KD cells. *CCL20* and *IL8* showed upregulation in BDH2-KD THP1 cells (Figure 5 and Appendix A).

## 3. Discussion

MDS is a hematological disease that is characterized by ineffective hematopoiesis [31,32]. Iron metabolism plays a crucial role in MDS. In addition, iron overload has an impact on leukemia transformation and survival [17,18,33], and is known to be related to GDF-11, GDF-15, and hepcidin. However, there are remaining unknown factors related to iron overload in MDS patients [13,14,15,34]. Studies have reported that BDH2 plays an important role in maintaining normal cytoplasmic and mitochondrial iron levels through LCN2-mediated iron trafficking activities [23,25]. In addition, lipopolysaccharide (LPS)-induced inflammation combined with endoplasmic reticulum (ER) stress leads to massive BDH2 downregulation, increased expression of ER stress markers, upregulated hepcidin expression, downregulated ferroprotein expression, iron retention in macrophages, and dysregulated cytokine release from macrophages [35].

Our study showed growth retardation of shRNA-BDH2 THP1 cells. This is consistent with our previous study, which concluded that there are possible mechanisms of inducing apoptosis [26], cell cycle arrest, and cell differentiation. In addition, increased apoptosis of BM precursors is a feature that is typically observed in the majority of MDS patients [36,37]. Analysis of the mechanisms involved in the increased rate of apoptosis in MDS showed the involvement of both an extrinsic pathway, the Fas/FasL pathway [38], and an intrinsic pathway through mitochondrial damage [39].

However, some MDS patients at advanced stages showed increased resistance to apoptosis [40], which may be due to the heterogeneity of cytogenetic abnormalities [41]. Apoptosis can be induced by reactive oxygen species (ROS) [42]. In our previous study, we showed that BDH2-KD THP1 cells are more sensitive to ROS-related apoptosis, which is associated with the anti-apoptotic protein survivin [26]. Perhaps the high BDH2 expression in MDS patients has an anti-apoptotic effect, and this may be one of the factors contributing to the more advanced disease status observed in these patients [43,44,45].

Another possible factor contributing to the growth rate might be differentiation. It has been observed that MDS patients with a lower expression level of CD11b have a lower survival rate and a higher risk of progression to AML [46]. In addition, CD15 is a marker of cell maturation, and an increase in the expression of CD15 has been reported in patients with MDS [46,47]. In this study, we noticed that there was an increase in the expression of CD11b but without a significant change in CD15 in shRNA-BDH2 THP1 cells. This suggests that a high expression of BDH2 cells is associated with a lower expression of CD11b, which is associated with a poor prognosis. Moreover, special stain studies for cell lines suggest that BDH2-KD cells have the potential to differentiate into cells with mature myeloid characteristics. Together, these results suggest that BDH2 may play a role in inhibiting differentiation.

The genes influenced by BDH2 may be involved in many pathways. Genes involved in cell metabolism, such as nicotinamide nucleotide transhydrogenase (NNT), which is involved in the tricarboxylic acid (TCA) cycle [48], and solute carrier 25 A1 (SLC25A1), which contributes to citrate transport and is involved in the wnt pathway [49,50,51], are downregulated in BDH2-KD THP1 cells. In addition, genes related to cell proliferation and differentiation are regulated by BDH2 and cause cell growth retardant in BDH2-KD cells. These genes include BBF2H7/CREB3L2, which is an ER-resident transmembrane transcription factor that responds to physiological ER stress and activates hedgehog signaling [52]; transforming growth factor-β (TGF-β) crosstalk between Smad- and MAP-kinase-dependent pathways [53] via TGF-β type I receptor [54]; and matrix metalloproteinase-9 (MMP-9), which is important for hematopoietic stem cell proliferation and differentiation [55].

Phosphoinositide-3-kinase catalytic delta polypeptide (*PIK3CD*) shows anti-apoptotic effects on all-trans-retinoic acid in acute promyelocytic leukemia cells [56] and is downregulated in BDH2-KD THP1 cells. In addition, *TNFSF13B,* which is related to B cell survival and maturation [57], is downregulated by knocking down BDH2. Moreover, other genes including interleukin (IL)-8 increase the apoptotic rate of leukemia cells [58]. Chemokine ligand (CCL) 20, a strong chemotactic for lymphocytes [59] that promotes growth of the prostate cancer cell line [60], and prolactin (PRL), which induces cell proliferation and differentiation via Src family tyrosine kinase [61] and the Jak2/Stat pathway [62], were upregulated during the down-regulation of BDH2. All of these downstream genes support the idea that BDH2-KD stimulates apoptosis and differentiation of cells. However, the results of gene expression by microarray only suggest a possible implication of the observed differentially expressed genes in the BDH2 pathway. The impacts of these genes in MDS need to be confirmed by their loss and gain functions in the future.

While the above results suggest the potential mechanism of action of BDH2, we also studied the impact of *BDH2* expression in MDS patients. The current study showed that the expression of *BDH2*mRNA was significantly elevated in the BM of MDS patients as compared with patients with normal BM. Interestingly, the patients with high *BDH2* expression showed a higher risk of leukemia progression, a poor IPSS-R risk, and shorter LFS when compared with patients with low *BDH2* expression. To assess whether higher *BDH2* expression was related to the increase in blast cells associated with leukemia progression, we analyzed BM samples from 13 patients from whom we had collected BM samples for at least two different stages of MDS. Only one MDS/MPN patient who underwent leukemia transformation, another who progressed from MDS/RAEB1 to MDS/RAEB2, another who progressed from MDS/MPN to leukemia transformation, and another who progressed from MDS/RAEB1 to MDS/RAEB2 to leukemia transformation showed increases in *BDH2*mRNA expression. This suggests that BDH2 expression is a prognostic factor for leukemia transformation but it is not strongly associated with blast number. While these results should be interpreted with caution due to the small sample size, they suggest that the number of blasts cells is not related to *BDH2*mRNA expression.

Ferritin is a storage form of iron [63]. Kikuchi et al. reported that serum ferritin levels at diagnosis could be associated with leukemia progression in MDS patients. In our study, plasma ferritin levels and *BDH2*mRNA expression showed a positive correlation. OS was significantly longer, while LFS showed a trend of being longer in the low-ferritin group than in the high-ferritin group. The plasma ferritin levels were positively related to *BDH2*mRNA levels but not *LCN2*mRNA levels. This may be attributed to the fact that we analyzed the total *LCN2*mRNA expression rather than two forms (holo- and apo-LCN2) separately. However, further studies are required to establish the role of BDH2 in iron trafficking, and subsequently in MDS.

The limitations of our study include the small number of MDS patients of each subtype who participated and the lack of a complete ferritin level before blood transfusion in our cohort. This was because we enrolled MDS patients from the years 2001 to 2012, and the ferritin level was not regularly checked in the early 2000s. However, owing to the long follow-up interval, comprehensive information such as survival was highly reliable. We also noticed that a mild decrease or lack of change in BHD2 mRNA expression under disease progression in several patients was in agreement with the absence of a general mechanism in MDS involving BDH2 expression.

In conclusion, *BDH2*mRNA expression is elevated in MDS patients and is correlated with the ferritin level. Patients with high *BDH2* expression showed a higher risk of leukemia progression, poorer IPSS-R scores, and shorter LFS when compared with patients with low *BDH2* expression. Therefore, BDH2 may be a factor in poor prognosis in MDS. Possible related mechanisms include anti-apoptosis, facilitation of the cell cycle, and inhibition differentiation through many pathways in malignant cells. Our study provides a foundation for further research on the roles of BDH2 and iron metabolism in the pathogenesis of MDS.

## 4. Materials and Methods

### 4.1. Patient Population

A total of 318 patients were enrolled at Kaohsiung Medical University, Chung-Ho Memorial Hospital, Taiwan, from 2001 to 2012. Of these, 199 were newly diagnosed patients with MDS. Only 187 patients with good mRNA quality were analyzed. The remaining 119 patients were diagnosed with de novo AML, including 3 patients with M0, 28 with M1, 52 with M2, 23 with M4, 7 with M5, 1 with M6, and 4 with M7, according to the World Health Organization (WHO) classification criteria [4]. Peripheral blood and bone marrow (BM) samples were collected from these patients. BM samples were also collected from 43 lymphoma patients with normal BM to be used as a control group. Normal BM was defined as normal BM findings proved by two hematologists and one pathologist, and no cytogenetic abnormalities. The protocol was approved by the Kaohsiung Medical University Hospital Institutional Review Board. All patients provided written informed consent.

### 4.2. RNA Interference-Mediated BDH2-KD in THP1 Cells and Cell Culture

The shRNA-BDH2 lentivirus was purchased from Sigma (St. Louis, MO, USA). The clones TRCN0000036735, TRCN0000036736, TRCN0000036738, and TRCN00000244979 were identified as shRNA-BDH2-1, shRNA-BDH2-2, shRNA-BDH2-3, and shRNA-BDH2-4, respectively. Naive THP1 acute myelomonocytic leukemia cells were transfected with lentivirus-expressing shRNAs and were selected for puromycin resistance (1 µg/mL). THP1 was cultured in RPMI complete medium. Puromycin was added for stress selection of THP1 cells infected with shRNA empty vector and shRNA-BDH2 lentivirus (Sigma, St. Louis, MO, USA, Appendix A). BDH2-KD efficiency was assessed by quantitative reverse-transcription polymerase chain reaction (qRT-PCR) and Western blot analyses.

### 4.3. qRT-PCR for mRNA Expression of BDH2 and LCN2

Total RNA was extracted from the BM samples of the enrolled patients and BDH2-KD THP1 cells using Trizol (Invitrogen, Life Technologies, Waltham, MA, USA). *BDH2*mRNA and *LCN2*mRNA expression was evaluated using specific forward, reverse primers, and the TaqMan^®^ probe [26]. Here, β-actin was used as an internal control (Appendix A).

### 4.4. Western Blot Analysis

Cells were starved by culturing in serum-free medium for 2 days. BDH2 expressions of nutrient-rich and -starved cells were detected by Western blot analysis, using primary antibodies specific for BDH2 (Sigma) and β-actin (Millipore Corporation, Billerica, MA, USA; Appendix A).

### 4.5. Flow Cytometry for Cell Differentiation and Cell Cycle Analysis

Flow cytometry was performed using a Gallios^™^ flow cytometer (BeckmanCoulter, Brea, California, USA). THP1 cells were initiated to differentiate along the monocytic lineage following exposure to 1.25-dihydroxyvitaminD3 (Vit D3; Sigma), as described previously [64]. Cell differentiation was detected using the following antibodies: CD11b-PE and CD15-FITC, CD14-PE and CD64-FITC, CD16-PE and CD14-FITC double stain, and immunoglobulin G1-PE and immunoglobulin G1-FITC double stain (Beckman Coulter).

For cell cycle analysis, THP1 cells were cultured in serum-free medium for 2 days to induce cell cycle arrest, and the cell cycle was restarted by supplementation with FBS (Appendix A). The cell cycle was analyzed by staining with propidium iodide (Beckman Coulter), and it was detected by flow cytometry (Appendix A).

### 4.6. Special Stains

THP1 cells were prepared by the cytospin technique. Liu staining, myeloperoxidase (MPO) staining, and nonspecific esterase (NSE) staining were performed for the analysis of cell morphologies and monocytic and myelocytic characteristics (Appendix A).

### 4.7. Immunohistochemistry (IHC)

Slides measuring 5 μm in thickness were cut from representative tissue blocks, IHC staining was performed using a monoclonal antibody against human BDH2 (sc-104197, Santa Cruz Biotechnology, Dallas, TX, USA), and counterstaining was done with hematoxylin (Appendix A).

### 4.8. mRNA Microarray Analysis

In order to find the target genes of BDH2, we used an mRNA microarray service (Phalanx Biotech Group, Inc., Taiwan) on shRNA-BDH2-3 THP1 cells and empty vector transfected THP1 cells. The array version was human OneArray microarray (HOA) 5.1. The number of genes subjected to advanced analysis was 29,189. Standard selection criteria to identify differentially expressed genes were established at log2 (fold change) ≥1 and *p* < 0.05.

### 4.9. Clinical Endpoints

The endpoint for the follow-up of patients without leukemia progression was the date of death or AML development, and for those who were lost to follow-up, it was the date of the last visit, denoted as “censored” data. OS was defined as the time measured from the date of initial diagnosis until the date of death of the patient; this was censored for patients who were alive at the last follow-up. LFS was defined as the time measured from the date of initial diagnosis to the date of leukemia progression or death from any cause.

### 4.10. Statistical Analysis

Statistical analyses were performed using SPSS 17.0 (IBM SPSS software, USA). Differences in *BDH2* and *LCN2* mRNA expression between patients with MDS subtypes, leukemia progression, de novo CN-AML, and normal BM were analyzed using analysis of variance (ANOVA). Age differences among patients with MDS, de novo AML, and normal BM were analyzed using ANOVA. Correlation regression analysis was used to assess the correlations among *BDH2* and *LCN2* expression and ferritin levels. The receiver operating characteristic (ROC) curve was used to estimate the cut-off point for *BDH2*—a delta *BDH2* of 10.10833. Patients with values above and below the cut-off point were defined as the BDH2^High^ and BDH2^Low^ expression groups, respectively (area under the curve (AUC) = 0.51; sensitivity = 57.6%; specificity = 35%). Time-to-event analysis involved estimating the probability that an event would occur at different time points. Chi-squared tests were used to analyze the risk of leukemia progression, as well as the IPSS-R [27] risk and MDS subtypes between the higher and lower *BDH2* expression groups. Kaplan–Meier curves were plotted to estimate survival.

## Figures and Tables

**Figure 1 ijms-21-03033-f001:**
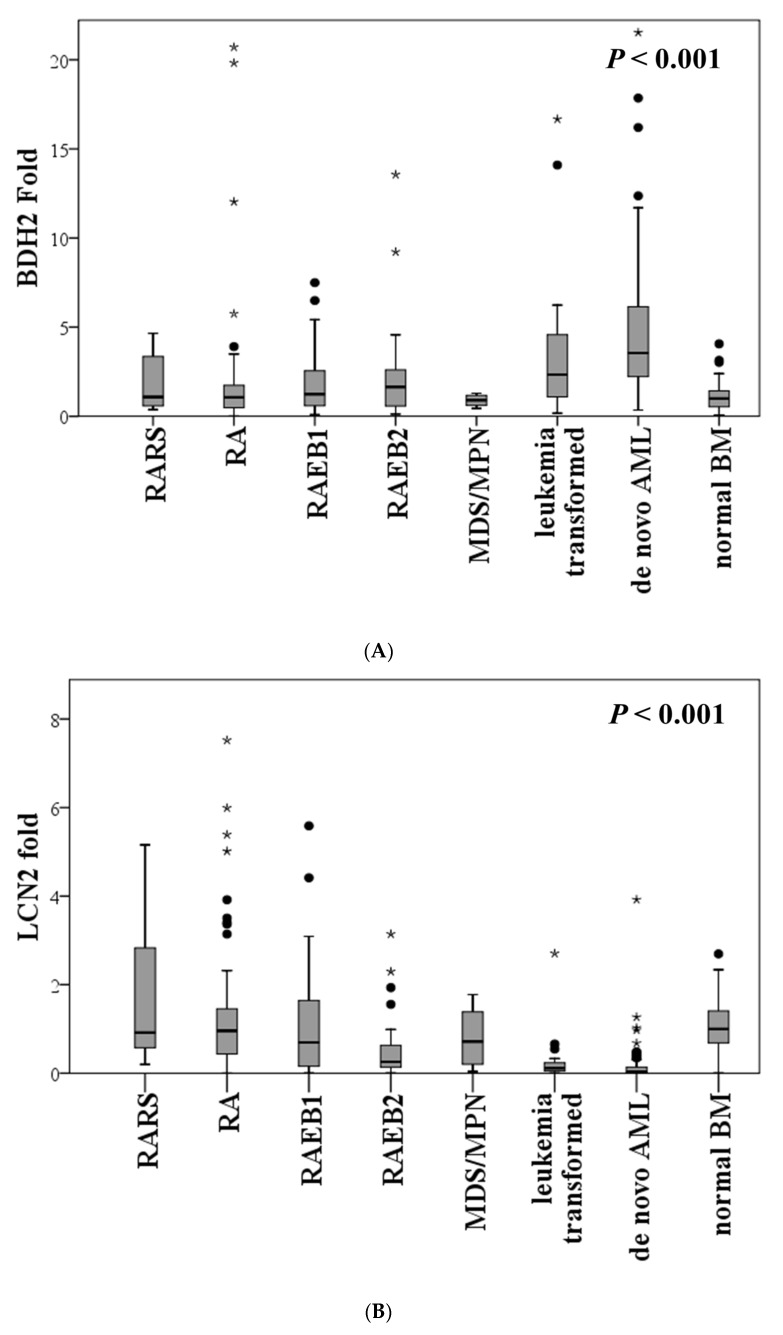
Expression levels of (**A**) *BDH2* and (**B**) *LCN2* mRNA in BM in MDS and control patients, including de novo CN-AML and normal BM. The expression levels of the *BDH2* and *LCN2* genes were normalized to the internal control β-actin to obtain the relative threshold cycle (ΔC_T_). BDH2, hydroxybutyrate dehydrogenase type 2; BM, bone marrow; CN-AML, cytogenetically normal acute myeloid leukemia; LCN2, lipocalin 2; MDS, myelodysplastic syndrome; RA, refractory anemia; RAEB, refractory anemia with excess blasts; RARS, refractory anemia with ringed sideroblasts.

**Figure 2 ijms-21-03033-f002:**
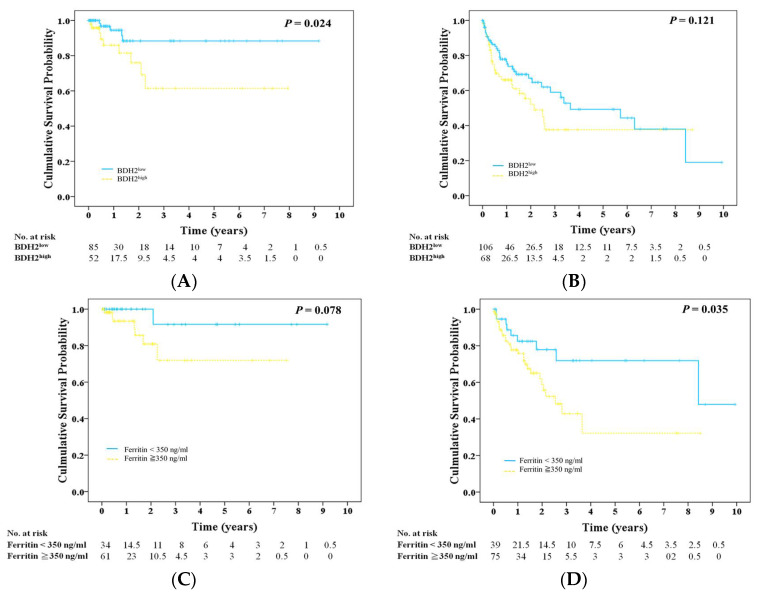
Survival rate in patients with MDS. (**A**) Leukemia-free survival and (**B**) overall survival in high and low *BDH2* expression groups. (**C**) Leukemia-free survival and (**D**) overall survival in patients with ferritin level less than, equal to, or more than 350 ng/mL. BDH2, hydroxybutyrate dehydrogenase type 2.

**Figure 3 ijms-21-03033-f003:**
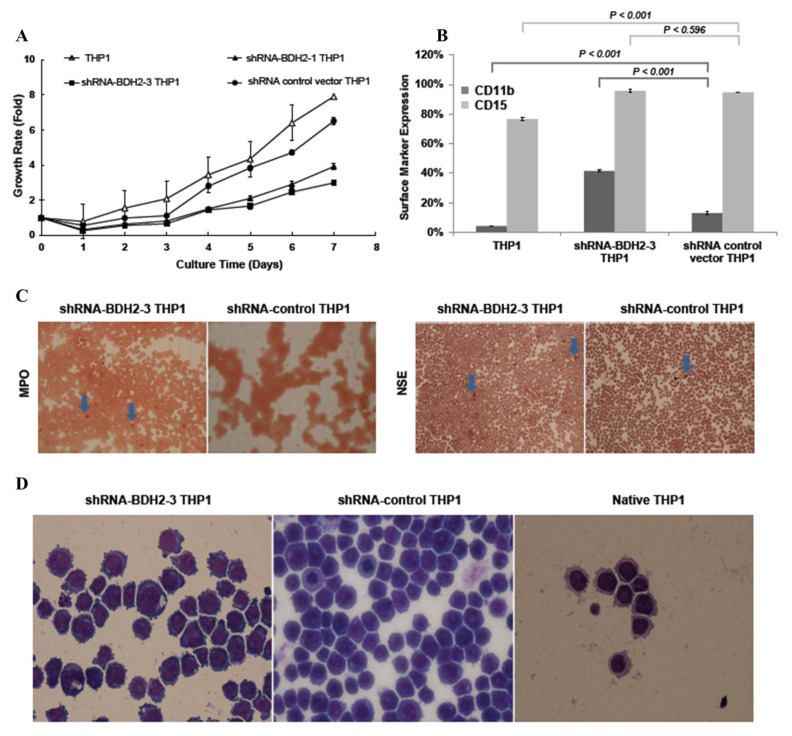
Cell growth rate and differentiation in BDH2 knockdown and native THP1 cells. Puromycin was used for stress selection for vector transfected cells. THP1 transfected with empty vector (shRNAc) was used as a control. (**A**) The cell growth rate in different THP1 cells. (**B**) The differentiation rate in THP1, BDH2 knockdown THP1, and shRNA control THP1 cell lines was measured by flow cytometry using the antibodies CD11b-PE and CD15-FITC. (**C**) NSE and MPO staining of BDH2 knockdown THP1 and control THP1 cells. The blue arrow indicates NSE-positive cells. (**D**) The morphology of THP1 after BDH2 knockdown. Lui stain, 40×.BDH2, hydroxybutyrate dehydrogenase type 2; CD, cluster of differentiation; MPO, myeloperoxidase; NSE, nonspecific esterase.

**Figure 4 ijms-21-03033-f004:**
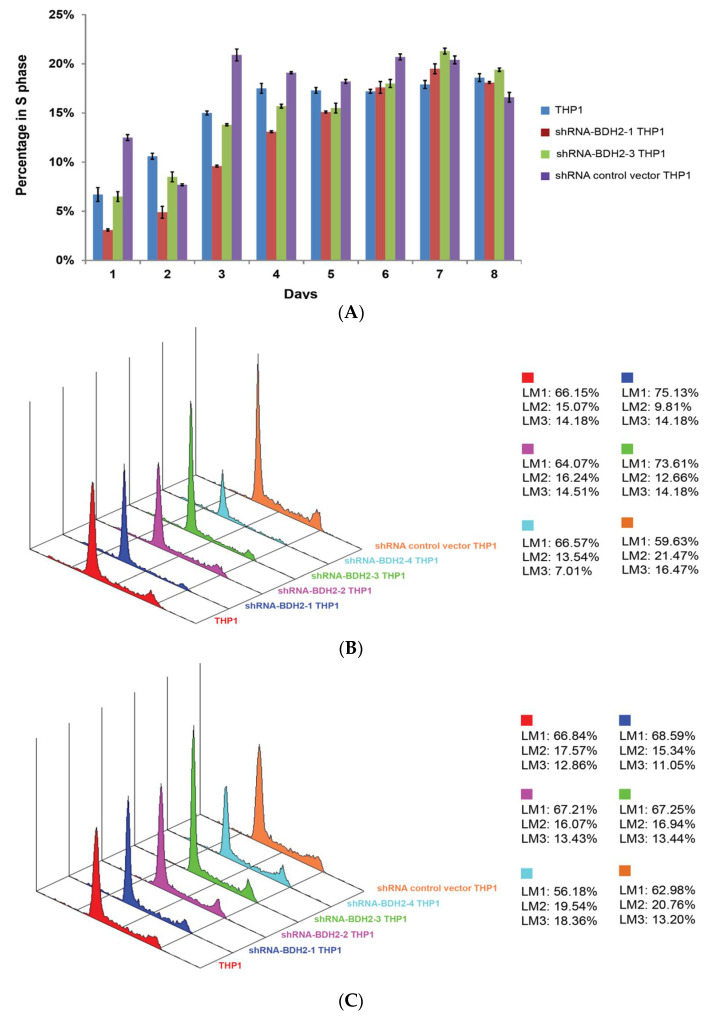
Cell cycle profile of native THP1 and shRNA control vector transfected THP1 cells from starving condition. (**A**) Cell cycle profile at the synthesis phase. (**B**) Analysis graphs on the third and (**C**) fourth days after re-culture with FBS. Native THP1 and shRNA control vector transfected THP1 cells entered S phase on day 3, while BDH2-KD THP1 cells started S phase on day 4. BDH2, hydroxybutyrate dehydrogenase type 2; FBS, fetal bovine serum. LM1: G0/G1; LM2: S; LM3: G2/M.

**Figure 5 ijms-21-03033-f005:**
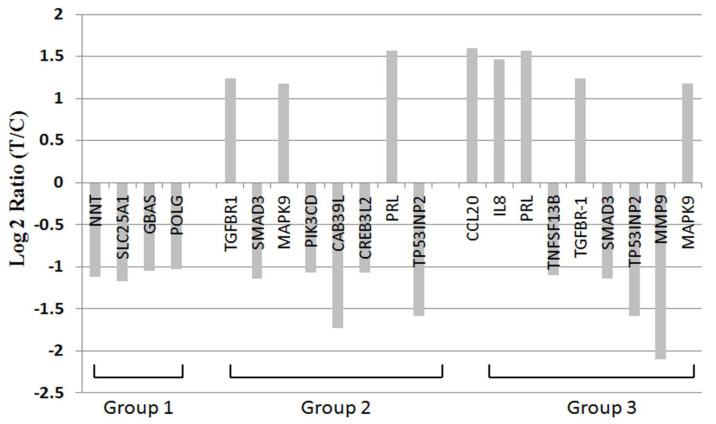
RNA microarray analysis of BDH2 targeting genes. Group 1: genes related to mitochondria; group 2: genes that have been reported as oncogenes or tumor suppressor genes; group 3: genes that function to control the survival, growth, differentiation, proliferation, and effector function of tissues and cells. All of those genes showed statistical significance with *p* < 0.05.

**Table 1 ijms-21-03033-t001:** Comparison of clinical manifestations and laboratory features in patients with MDS in low and high BDH2 expression groups *.

Variables	Total (*n* = 186)	Low BDH2 Expression (*n* = 114)	High BDH2 Expression (*n* = 73)	*P*
**Sex ^†^**				0.444
Male	119	75 (65.79%)	44 (60.27%)	
Female	68	39 (34.21%)	29 (39.73%)	
**Age (years) ^‡^**		65 (19–87)	66 (35–87)	0.976
**Laboratory data ^‡^**				
WBC, μL^−1^	3990 (1100–56,000)	3945 (1220–34,000)	4100 (1100–56,000)	0.144
Hb, g/dL	8.4 (4.3–14.7)	8.55 (4.3–13.5)	8.4 (5.6–14.7)	0.720
MCV, fl	99.2 (52–122.7)	98.5 (70–122.7)	99.2 (52–117)	0.812
Platelets, ×1000/μL	79,000 (8000–456,000)	94,000 (8000–456,000)	66,000 (12,000–346,000)	0.051
Ferritin, ng/mL ^||^	555.75 (62–5912)	423.3 (8.1–5614)	1003.6 (133.1–5916)	0.02 **
**Type ^†^**				0.003 **
RA	65	47 (35.09%)	18 (24.66%)	0.027 **
RARS	14	10 (8.77%)	4 (5.48%)	0.336
RAEB-1	38	21 (18.42%)	17 (23.29%)	0.420
RAEB-2	36	17 (14.91%)	19 (26.03%)	0.060
MDS/MPN	12	11 (9.65%)	1 (1.37%)	0.024 **
Leukemia change	22	8(7.02%)	14 (19.18%)	0.012 **
**IPSS-R score ^§†^**				0.004 **
Very low	6	5 (4.39%)	1 (1.37%)	0.254
Low	53	40 (35.09%)	13 (17.81%)	0.011 **
Int	43	29(25.44%)	14 (19.18%)	0.321
High	32	13 (11.40%)	19 (26.03%)	0.017 **
Very high	47	22 (19.30%)	25 (34.25%)	0.022 **
**Leukemia progression ^§†^**		4 (3.77%)	9 (15.25%)	0.017 **

* The median value of BDH2 expression in the total population of 10.10833 was used as the cut-off level to define the low and high expression groups. ^†^ Number of patients (%). ^‡^ Median (range). ^§^ Patients with more than 30% blasts in bone marrow were excluded. ^||^ Only 109 patients had ferritin measured at diagnosis. ** Statistically significant (*p* < 0.05). BDH2, hydroxybutyrate dehydrogenase type 2; Hb, hemoglobin; Int, intermediate; IPSS-R, Revised International Prognostic Scoring System; MCV, mean corpuscular volume; MDS, myelodysplastic syndrome; MPN, myeloproliferative neoplasm; RA, refractory anemia; RAEB, refractory anemia with excess blasts; RARS, refractory anemia with ringed sideroblasts; WBC, white blood cells.

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
