# Peer review of "The Effects of Human BDH2 on the Cell Cycle, Differentiation, and Apoptosis and Associations with Leukemia Transformation in Myelodysplastic Syndrome"

_ijms, 2020, doi:10.3390/ijms21093033_

Round 1
Reviewer 1 Report
Yang et al show that BDH2 is involved in cell cycle arrest and inhibition of differentiation in malignant cells. High BDH2 expression in MDS patients suggest that it could act as a poor prognostic factor. This study provides a foundation for further research on the role of BDH2 and iron metabolism in the pathogenesis of MDS. I liked the paper and therefore I recommend it for publication, pending professional English proofing.
Author Response
Dear Reviewer:
Thank you very much for your commend. I already re-revise the English.
Reviewer 2 Report
Reviewer comments:
The authors observed that, BDH2mRNA expression is elevated in MDS patients, with correlation with ferritin level. Patients with high BDH2 expression showed higher risk for leukemia progression, poor IPSS-R score, and shorter LFS, compared with patients with low BDH2 expression. Therefore, BDH2 expression may be a poor prognostic factor in MDS, with the possible mechanism as an anti-apoptosis factor, facilitating cell cycle, and inhibits differentiation in malignant cells through many pathways. They observed that BDH2mRNA expression levels were higher in high- and very high-risk MDS patients than in low-risk patients.
I have nonetheless some concerns:
The authors didn’t mentioned which BM cells they used, are they used purified cells, mononuclear cells, red blood cells?
How the authors can explain that among the 13 patients analyzed using BM samples preserved at different stages of MDS only four patients showed increase in BDH2 mRNA expression under progress of the decease.
The fact that authors showed mild decrease or no change in BHD2 mRNA expression under disease progression is in agreement with the absence of a general mechanism in MDS involving BDH2 expression. The authors should add comments about that.
It is not clear how Plasma ferritin levels were has no significant linear relationship with the mRNA expression levels of BDH2 and LCN2 but serum ferritin levels were higher in patients belonging to the high BDH2mRNA expression group.
In page 9, the title 2.5.BDH2-KD THP1 Showed Slow Growth Rate, Related to Cell Differentiation and (the end of sentence is missing)
The use of the acute myelomocytic leukemia THP1 cell line must be argued, and more detail on the origin and the selection of these cells should be provided. The authors should give complete explanations in the Introduction, including the history of these cell line were used as common MDS modeled tools.
Although authors used 4 clones types of shRNA-BDH2, they detailed only the results of two of them, Please discuss or show additional experiments with the other clones to complete the conclusion.
The up- regulation of CCL20 and IL8 in BDH2-KD THP1 cells should be discuss. In addition, the results of genes expression by microarray only suggest a possible implication of the observed differentially expressed genes in the BDH2 pathway. These results should be confirmed by lost and gain of function before than the authors can conclude to their implication in the MDS disease.
Overall, it is a clear manuscript but unfortunately many sentences should be reviewed by a person fluent in English to facilitate understanding. The article is well documented and describes the experimental procedures proposed by the authors. Therefore, the flow of ideas in this document once corrected can be reconsidered.
Author Response
Thank you very much for your commend. I already re-revise the English. Please see the attached file
